# A Randomized Controlled Trial to Assess the Feasibility and Practicability of an Oatmeal Intervention in Individuals with Type 2 Diabetes: A Pilot Study in the Outpatient Sector

**DOI:** 10.3390/jcm13175126

**Published:** 2024-08-29

**Authors:** Michél Fiedler, Nicolle Müller, Christof Kloos, Guido Kramer, Christiane Kellner, Sebastian Schmidt, Gunter Wolf, Nadine Kuniß

**Affiliations:** 1Department of Internal Medicine III, Jena University Hospital, 07747 Jena, Germany; michel.fiedler@uni-jena.de (M.F.); nicolle.mueller@med.uni-jena.de (N.M.); christof.kloos@med.uni-jena.de (C.K.); guido.kramer@med.uni-jena.de (G.K.); christiane.kellner@med.uni-jena.de (C.K.); sebastian.schmidt@med.uni-jena.de (S.S.); gunter.wolf@med.uni-jena.de (G.W.); 2Outpatient Healthcare Center MED:ON MVZ, 99096 Erfurt, Germany

**Keywords:** oatmeal intervention, oats, diabetes, insulin resistance, blood glucose, pilot study

## Abstract

**Background/Objectives**: The aim of this study was to investigate the feasibility and practicability of repeated three-day sequences of a hypocaloric oat-based nutrition intervention (OI) in insulin-treated outpatients with type 2 diabetes and severe insulin resistance. **Methods**: A randomized, two-armed pilot study was conducted with three months of intervention and three months follow-up with 17 participants with insulin resistance (≥1 IU/kg body weight). Group A (*n* = 10) performed one sequence of OI; Group B (*n* = 7) performed two sequences monthly. A sequence was 3 consecutive days of oat consumption with approximately 800 kcal/d. The main objective was to assess feasibility (≥70% completers) and practicability regarding performance aspects. Biomedical parameters such as HbA1 c were observed. To evaluate the state of health, a standardized questionnaire was used (EQ-5 D). **Results**: OI was feasible (13/17 completer participants (76.5%): 70.0% Group A, 85.7% Group B). Individually perceived practicability was reported as good by 10/16 participants (62.5%). Total insulin dosage decreased from 138 ± 35 IU at baseline to 126 ± 42 IU after OI (*p* = 0.04) and 127 ± 42 IU after follow-up (*p* = 0.05). HbA1 c was lower after OI (−0.3 ± 0.1%; *p* = 0.01) in all participants. Participants in Group B tended to have greater reductions in insulin (Δ−19 IU vs. Δ−4 IU; *p* = 0.42) and weight loss (Δ−2.8 kg vs. Δ−0.2 kg; *p* = 0.65) after follow-up. Severe hypoglycemia was not observed. EQ-5 D increase not significantly after follow-up (57.2 ± 24.0% vs. 64.7 ± 21.5%; *p* = 0.21). **Conclusions**: The feasibility and practicability of OI in outpatients were demonstrated. OI frequency appears to correlate with insulin reduction and weight loss. Proper insulin dose adaptation during OI is necessary. Presumably, repeated OIs are required for substantial beneficial metabolic effects.

## 1. Introduction

Being overweight is a relevant contributor to the development and progression of type 2 diabetes mellitus (T2 DM) and leads to decreased insulin response and subsequently to diminished pancreatic insulin secretion later in the course of the disease [1]. Furthermore, overweight is a contributor to insulin resistance (IR), resulting in deteriorated glucose disposal and hyperinsulinemia, often accompanied by hypertension and dyslipidemia [2]. In some individuals, high doses of insulin can be required on top of other antidiabetic drugs like metformin to maintain glucose control [3]. However, even with the maximum non-insulin and insulin therapy, some patients fail to achieve glycemic control, and treatment remains challenging, thus increasing the risk of diabetes-associated diseases [4]. In addition, high doses of insulin are associated with the risk of hypoglycemia and provoke adverse cardiac events as well as further increases in body weight [5,6].

As the cornerstone of IR management, lifestyle modifications including increased physical activity and dietary changes are recommended [7,8]. Oat consumption is considered a useful form of dietary intervention on poorly controlled T2 DM [9]. Research has suggested that ingredients in oats, mainly beta glucan as a source of soluble fibers, may contribute to lowering glucose and cholesterol [10,11,12].

Storz et al. conducted a review evaluating the potential health benefits and therapeutic effects of oats and showed favorable outcomes in three inpatient interventions [13]. An oat-based diet was followed on two consecutive days in individuals with inadequate glycemic control during an inpatient stay, leading to improvements in mean blood glucose and a reduction in daily insulin dose up to 40% [14] and 31% [15], respectively. In a crossover inpatient study from 2019, Delgado et al. also reported a significant decrease in insulin demand comparing oatmeal treatment with a diabetes-adapted diet in 15 patients with uncontrolled T2 DM; lower HbA1 c levels were confirmed up to four weeks after the end of the intervention [16]. This led to the assumption that a periodically repeated oatmeal intervention (OI) may have a favorable impact on metabolism and reduce IR, leading to better diabetes control [9,13]. As a special regime to overcome IR in clinical settings, *Hafertage* (oat-days) are mentioned as a therapeutic tool in the German 2023 Clinical Practice Recommendations for Nutrition in T2 DM therapy [17], although clear evidence is lacking. Furthermore, no application in an outpatient setting has been hitherto studied. This study examines the feasibility and practicability of an oat-based dietary intervention in the daily routine of obese outpatients with T2 DM and insulin resistance. Additionally, this study examines if OI can ameliorate IR and glycemic control in these patients.

## 2. Materials and Methods

This trial was designed as a clinical, randomized controlled, two-armed pilot study. This trial took place at the outpatient Department of Endocrinology and Metabolic Diseases at the University Hospital Jena (Germany) between September 2021 and February 2023. The screening of participants was performed at three institutions: (1) Division of Endocrinology and Metabolic Diseases of the Jena University Hospital, (2) Practice for Diabetology in Jena and (3) Practice for Diabetology in Erfurt. Patients were enrolled according to the following criteria: those with insulin-treated T2 DM with severe IR, defined as an insulin requirement of ≥1 IU/kg body weight, HbA1 c >7.0% and the ability to speak and understand the German language. Due to an insufficient number of participants until September 2022, in an amendment, the baseline HbA1 c was reduced from 7.5 to 7.0%. Exclusion criteria included any previous oat-based interventions during the last 12 months, eating disorders, dementia, arrhythmia and pregnancy/ breast feeding. This trial was approved by the ethics committee of the Jena University Hospital (number 2020-1905_2-BO) and was performed according to the principles of the Declaration of Helsinki and Good Clinical Practice. Retrospectively, this trial was entered at the German Register of Clinical Trials (number DRKS00033580). Individuals were informed about this study and gave written consent (Figure 1).

### 2.1. Intervention

Participants were randomized 1:1 to Group A or B. Randomization was undertaken independently at the Institute of Medical Statistics, Computer Sciences, and Documentation, Jena University Hospital. Group A was instructed to perform one sequence of three consecutive OI days per month, whereas Group B performed two sequences. The study duration was six months in total, three months of OI followed by a period of three months of follow-up. Apart from OI, no further nutritional specifications were given.

At study entry, the participants were trained in OI meal preparation, and recipes were handed out. Study participants were instructed to perform OI as follows: one oat-day included three self-prepared meals with 60 g of rolled oats per portion. Different recipes for the two taste options sweet and savory were given: The sweet version of oats (270 kcal, 41 g carbohydrates (CHO), 9 g protein, 7 g fat; Appendix A was enriched with 200 mL unsweetened almond drink (2.3% almonds) (i.e., 50 g CHO). To make meals more palatable, fruit according to personal preferences could be added, containing 6 g CHO, e.g., 50 g apple or 30 g banana or 100 g strawberries. In the savory version (261 kcal, 40 g carbohydrates (CHO), 10 g protein, 5 g fat; Appendix A), oats were prepared in 200 mL of broth with 150 g vegetables (containing 4.5 g CHO) and flavored with herbs ad libitum (i.e., 40 g CHO). Participants could choose which type of preparation they preferred. The total daily intake was 180 g oats (containing 120 g CHO) and approx. 800 kcal (Appendix A). During OI days, no further food intake was allowed, unless they were carbohydrates to counteract hypoglycemia. An adequate calorie-free fluid intake (1,5–2 L/d) was recommended. These dietary recommendations were the same for all study patients regardless of body weight and activity level.

At study entry, the baseline dosage of short- and long-time-acting insulin were documented and reduced up to 40% during OI days to prevent hypoglycemia, based on findings from previously published papers with inpatients [14,15,16]. All participants received a glucose sensor (CGM “Freestyle Libre 3”) for continuous measurement and were recommended to frequently check their glucose levels and self-adjust insulin dosage if necessary. In addition, an information sheet was handed out. Telephone and on-site consultations to adjust insulin dosage were performed throughout the whole study according to study protocol (Figure 1).

To ensure a correct dietary procedure, participants were asked to fill out two protocols around OI: one regarding the served oatmeal and its performance aspects (e.g., preparation, taste, satiety and possible protocol deviations like additional food intake) and one regarding glycemic control (e.g., insulin doses, hypoglycemia), starting three days before and three days after each OI sequence. Additionally, the participants were coached via repeated visits and telephone calls. Hypoglycemia was defined as values below 4 mmol/L or the presence of symptoms [18]. After the 6-month study period, participants evaluated the practicability and acceptance of the intervention.

### 2.2. Outcomes

The primary end point was to evaluate the feasibility and practicability of OI during the 3-month intervention period. Feasibility was assumed to be present if 70% or more of the study cohort successfully performed the intervention. As a qualitative analysis, a structured interview was conducted about performance aspects in participants’ everyday life, such as time effort to obtain ingredients, taste, preparation and satiation. Furthermore, participants were required to document protocol deviations. Food intake because of hypoglycemia was not considered as such. After intervention [t(1)] and after follow-up [t(2)], practicability was assessed via a structured protocol about the frequency of OI, implementation in daily life and its social acceptance.

Secondary end points were changes in insulin dosage, HbA1 c, frequency of hypoglycemia, body weight, blood pressure and satisfaction and state of health through the intervention. Patients’ characteristics and metabolic parameters were drawn from the electronic patient record EMIL^®^. Treatment was evaluated with the DTSQ questionnaire with items to rate current diabetes therapy and the frequency of hyper- or hypoglycemic events [19]. Baseline satisfaction was calculated via a 0–36 scale using the DTSQs (36 highest satisfaction). At t(1) and t(2), an adapted version (DTSQc) was handed out to evaluate changes in satisfaction scored from −18 to +18. A positive score indicates an improvement in satisfaction, while a negative score indicates a deterioration. The same approach was used to detect changes in the frequency of hyperglycemia or hypoglycemia with a score ranging from −3 to +3 compared to t(0). The individually perceived state of health was analyzed with the EQ-5 D questionnaire ranging from 0 to 100% (100% best possible health) [20].

### 2.3. Statistical Analysis

Continuous data are expressed as the mean ± standard deviation (SD). To compare two groups, an unpaired *t*-test was used for continuous variables. A paired *t*-test was used to analyze differences between baseline and follow-up in both groups. Questionnaires were completed at study entrance, at the end of the 3-month intervention and follow-up. A sample test was conducted based on a score of 0 for no changes being made. A *p*-value < 0.05 was considered statistically significant. The analysis was conducted under the principle of intention-to-treat using IBM SPSS Statistics 27. For statistical analysis, we performed the last-observation-carried-forward (LOCF) method to fill gaps in data for biomedical outcomes. Thereby, 17 participants were analyzed for biomedical outcomes, while 16 participants were included for the analysis of practicability protocol and questionnaires.

## 3. Results

In total, 350 patients were screened for eligibility. A total of 42 patients met the inclusion criteria, and 17 patients agreed to participate in this study: Group A (*n* = 10) and Group B (*n* = 7). Three participants in Group A and one in Group B discontinued this study during intervention but consented to follow-up. One of these participants who discontinued this study at week five died at week twelve due to an underlying medical condition unrelated to the study intervention (Figure 1).

The mean age of the participants was 70.7 (±7.1) years with a diabetes duration of 21.5 (±8.7) years (58.8% male). The mean insulin dosage at baseline was 138 (±35) IU (Table 1).

### 3.1. Feasibility of Intervention

A total of 13 out of 17 individuals (76.5%) completed the intervention period with no difference concerning the intervention arms (Group A: 7/10 participants (70.0%); Group B: 6/7 (85.7%); *p* = 0.60). Two out of four patients left the intervention due to OI-related reasons (unsatisfaction with OI [one], laboriousness for self-preparation at home [one]).

### 3.2. Practicability of Intervention

Most participants reported that the effort for preparing the meals was adequate and the taste was good overall, but satiety was judged as insufficient (Table 2). Approximately half of the participants (Group A: 5/9 (55.6%), Group B: 3/7 (42.9%)) stated that three consecutive OI days were too difficult to endure (*p* = 1.00). Considering the repetitions, 7/9 (77.8%) in Group A considered OI appropriate, and 2/9 (22.2%) considered one sequence per month not enough; in Group B, 5/7 (71.4%) considered two sequences per month appropriate (*p* = 1.00). OI could be more often integrated (non-significantly) in daily life in Group B (6/7; 85.7%) than Group A (4/9; 44.4%; *p* = 0.15). The social acceptance of OI was high in both groups, and 13 out of 16 participants (81.3%) reported that they would repeat OI.

Occasionally, minor protocol deviations occurred in all sequences, leading to additional food intake. The reasons were mainly hunger or need for variety in taste. Additional carbohydrate ingestion varied from 5 g to 110 g. On average, an additional 40 (±30) g of CHO was consumed during the intervention.

### 3.3. Biomedical Outcomes

The total insulin per day decreased statistically significantly from t(0) to t(1) [−12 IU, *p* = 0.04] and from t(0) to t(2)) [−11, *p* = 0.05] (Table 3). In Group A, total insulin decreased non-significantly from t(0) to t(1) [−11 IU, *p* = 0.12] and from t(0) to t(2)) [−4 IU, *p* = 0.05]; in Group B, total insulin decreased by −11 IU [t(0) vs. t(1), *p* = 0.25] as well as −19 IU [t(0) vs. t(2), *p* = 0.04]. Over all participants, short-acting insulin per day decreased significantly by −10 IU at t(1) [*p* = 0.03] and t(2) [*p* = 0.05] compared to baseline. This was due to a marked decrease in Group B (−19 IU; *p* = 0.03), whereas in Group A, insulin dosage did not change. No significant changes occurred for long-acting insulin. The correlation between the mean blood glucose and insulin demand during intervention days showed a statistically significant reduction in applied insulin dosage when compared to baseline (Appendix A).

At t(1), HbA1 c improved from 8.1(±0.5)% to 7.8(±0.6)% [*p* = 0.01] but returned to the baseline value (8.1(±0.6)%, [*p* = 0.71]) at t(2). The effect in the whole group was again due to the marked improvement in Group B (HbA1 c −0.5% at t(1); [*p* = 0.004])*,* whereas HbA1 c in Group A remained unchanged. The same pattern was present for body weight which decreased in the whole group from 103.9(±20.3) kg to 102.6(±20.3) kg; *p* = 0.04 at *t*(2) but again only due to the participants in Group B (−2.8 kg, *p* = 0.02). In addition, systolic blood pressure tended to decrease from 149(±20) mmHg to 144(±24) mmHg at t(2) [*p* = 0.48], mainly attributable to effects in Group B (−12 mmHg, *p* = 0.34) rather than Group A (+1 mmHg, *p* = 0.78). Changes in diastolic blood pressure showed a comparable pattern, Figure 2.

Hypoglycemic events particularly occurred during the first sequence of OI (Group A (*n* = 2) and Group B (*n* = 2)).

### 3.4. Treatment Satisfaction and State of Health

The mean baseline DTSQ score was 28(±4.4). The treatment satisfaction (DTSQc) of all 16 analyzed participants increased markedly by +8.7(±4.7) at t(1) [*p* < 0.001] and continued to rise afterwards (t(2): +10.8(±5.8); [*p* < 0.001]). At study entrance, the perceived frequency of hyperglycemia was 3.8(±1.6) and slightly increased at t(1) (DTSQc score +0.3(±5.8); [*p* < 0.001]) and decreased at t(2) (DTSQc score −0.1(±1.2); [*p* < 0.001]). The frequency of hypoglycemia was not changed throughout this study (Appendix A). The total score in the state of health improved non-significantly from 57(±24)% at t(0) to 65(±22)% at t(2) [*p* = 0.21] (Appendix A).

Daily carbohydrate intake remained unchanged at t(2) compared to t(0) (Appendix A).

### 3.5. Adverse Events

Over the entire study, hypoglycemic events occurred in 6 out of 17 participants (35.3%; A: *n* = 3; B: *n* = 3) with a mean average of 3.6 (±0.3) mmol/L, without severe hypoglycemia. During the OI sequence, 3/16 participants (18.8%) reported to have suffered from obstipation, 2/16 (12.5%) a feeling of fullness, 2/16 (12.5%) reported nausea or diarrhea and 1/16 (6.3%) experienced a headache. These events were temporary and diminished as the intervention sequence continued.

## 4. Discussion

Hypocaloric interventions are known to have beneficial effects in individuals with T2 DM which may be intensified by certain nutrition ingredients, e.g., oatmeal [9]. It is unknown whether patients integrate and conduct such an intervention and in which way this intervention is best fitted in patients’ daily life. In this two-armed, randomized controlled pilot study, we demonstrated that a repeated OI is feasible (76.5% successful study population) in patients with severe IR on an outpatient basis.

In addition, the majority of participants evaluated the intervention as practical in daily life. Treatment satisfaction was high. A total of 50.0% considered three consecutive days of OI to be too long; the majority would prefer two days instead. Most participants reported that the effort in preparing meals was adequate, and the taste was good overall, but satiety was judged as insufficient. The social acceptance of OI was high in both groups. A total of 81.3% reported that they would repeat OI. Only two patients left the intervention due to OI-related reasons (unsatisfaction with OI as well as laboriousness for self-preparation at home).

Overall, the occurrence of adverse events was relatively low. During the OI sequence, two of five participants reported to have suffered from obstipation, 12.5% a feeling of fullness, 12.5% reported nausea/diarrhea and 6.3% experienced a headache.

Regarding biomedical outcomes, our findings are consistent with previous clinical studies performed during a hospital stay [14,15,16]. Daily insulin dose during OI decreased significantly, and glycemic control remained stable. Lammert et al. reported a lasting reduction in insulin demand by approximately 40% after 4 weeks of follow-up from a hospital stay with only two consecutive OI days (from 145 to 83 IU/d) [14]. In our study, we only found a total reduction of 11 IU (~8%) after the 3-month follow-up. Lammert et al. discussed that the hypocaloric nutrition regime and decreased intake of saturated fats could be possible explanations [14]. However, we followed the same approach and did not see such a long-lasting impact. This may be attributed to our outpatient study design compared to a hospital setting. Our study was also conceived to address insulin need and IR throughout a hypocaloric intervention with oat-days as proposed by the guideline of the German diabetes association [17]. Participants in Group B obtained marked weight loss, displayed markedly reduced daily insulin doses and blood pressure, which implies a connection regarding the benefit. This may be due to the more intensive intervention. Although insulin levels remained high and IR was not overcome during the intervention, insulin reduction may be more pronounced if OI is performed repeatedly over a longer period of time. This finding should be verified in a randomized study with a longer interventional period.

Research has shown that CGM improves glycemic control by extending time in range and providing detailed information about glucose variability [21,22]. Especially during the beginning of this study and the first meal interventions, CGM is useful for monitoring blood glucose and makes insulin dose adaptation easier and thus less risky for the patient.

Despite intensive screening, we had a shortfall of individuals meeting both the inclusion criteria of insulin resistance and increased HbA1 c. This unexpected fact may be attributed to recent advances in antidiabetic therapy apart from metformin namely GLP-1-analoga and SGLT-2 inhibitors [23]. In a sub-analysis, we found a high burden of comorbidity in study participants, which becomes apparent considering the mean average of 13 medications per day and person (Appendix A). Albeit the negative selection of patients, 81.3% of study participants would be up to repeat OI, indicating that study participation was tolerable.

### Strengths and Limitations of This Study

To the best of our knowledge, the present study shows, for the first time, the feasibility and practicability of OI as a proof-of-principle to be performed in an outpatient and real-world setting with a well-characterized population. During the study period, multiple consultations were performed, resulting in no relevant hypoglycemia and strengthened doctor–patient relationships. In addition, the benefits of OI have been shown to provide a simple tool to enhance metabolic control, thereby encouraging health-seeking behavior to address participants’ nutritional imbalances.

A major limitation is the small sample size and the lack of a control group. Whether oats (and its specific fiber) as such or the intervention-induced weight loss are causal for the observed reduction in insulin dosage and improved glycemic control is still unclear. OI should be compared with another hypocaloric diet, such as vegetables or rice in further studies. Nevertheless, it is important to note that this was not the primary outcome of this pilot study. Furthermore, the outpatient design and home-based intervention did not allow for the supervision of protocol adherence. However, we interviewed patients about protocol discrepancies during the intervention and requested that they document them.

## 5. Conclusions

OI is feasible in outpatients with high insulin doses. The intervention leads to reduced insulin requirement and improved glycemic control. To avert hypoglycemia, supervised insulin dose adjustment is crucial. Repeated OI applications may produce lasting and substantial effects. The findings of this pilot study should result in further large-scale studies to assess the sustained metabolic effects of OI as a possible non-pharmacological treatment option in T2 DM patients with insulin therapy.

## Figures and Tables

**Figure 1 jcm-13-05126-f001:**
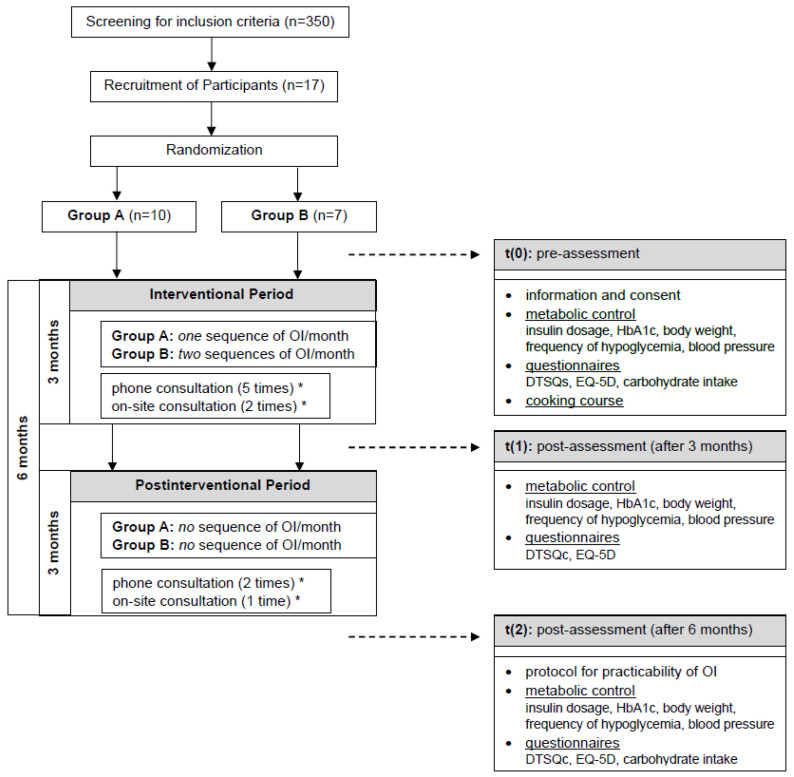
Flow chart of recruitment and study procedure. Sequence of OI = three consecutive days. * adjustment of insulin dosage, checking for hypoglycemia, checking for adverse events on every consultation.

**Figure 2 jcm-13-05126-f002:**
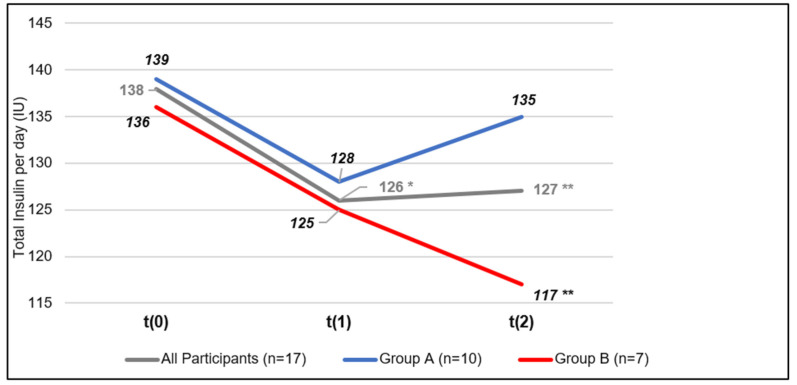
Course of total insulin per day at t(0): study entry, t(1): after intervention, t(2): after follow-up. * *p* < 0.05 t(0) vs. t(1); ** *p* < 0.05 t(0) vs. t(2).

**Table 1 jcm-13-05126-t001:** Baseline characteristics of study population.

	All Participants (*n* = 17)	Group A (*n* = 10)	Group B (*n* = 7)	*p*-Value (A vs. B)
Sex Male Female				0.35
	10 (58.8%)	7 (70.0%)	3 (42.9%)	
	7 (41.2%)	3 (30.0%	4 (57.1%)	
Age (years)	70.7 ± 7.1	69.3 ± 7.8	72.7 ± 5.8	0.35
Diabetes duration (years)	21.5 ± 8.7	22.4 ± 10.3	20.3 ± 6.4	0.64
Insulin dosage (IU)				
Total	138 ± 35	139 ± 34	136 ± 40	0.85
Short-acting	105 ± 37	107 ± 34	102 ± 43	0.77
Long-acting	36 ± 27	32 ± 21	41 ± 35	0.53
Form of therapy				0.42
ICT	14	8	6	
CT	1	0	1	
SIT	1	1	0	
Insulin pump	1	1	0	
HbA1 c (%)	8.1 ± 0.5	8.0 ± 0.5	8.1 ± 0.6	0.86
HbA1 c (mmol/mol)	65.03 ± (−18.03)	63.93 ± (−18.03)	65.03 ± (−16.94)	0.86
Body weight (kg)	103.9 ± 20.3	104.8 ± 18.2	102.5 ± 24.6	0.82
BMI (kg/m^2^)	35.9 ± 5.5	35.6 ± 4.9	36.4 ± 6.7	0.78
Systolic blood pressure (mmHg)	149 ± 20	146 ± 17	152 ± 25	0.55
Diastolic blood pressure (mmHg)	85 ± 15	82 ± 11	89 ± 19	0.35

**Table 2 jcm-13-05126-t002:** Protocol for practicability measured after 6 months of follow-up t(2), (*n* = 16).

**(1) Number of Oat-Days (3 Consecutive Days)**				
	too many	appropriate	too few	
Group A	5	4	-	*n* = 9
Group B	3	4	-	*n* = 7
**(2) Number of Sequences (one sequence vs. two sequences per month)**				
	too many	appropriate	too few	
Group A	-	7	2	*n* = 9
Group B	1	6	-	*n* = 7
**(3) Implementation in Daily Life**				
	good	neither good nor bad	bad	
Group A	4	4	1	*n* = 9
Group B	6	1	-	*n* = 7
**(4) Social Acceptance**				
	“I would repeat OI again”		“I would not repeat OI again”	
Group A	7		2	*n* = 9
Group B	6		1	*n* = 7

**Table 3 jcm-13-05126-t003:** Secondary outcomes after study intervention (t(1)) and follow-up (t(2)), all patients and group differences.

	All Participants (*n* = 17)	*p*-Value t(0) vs. t(1) t(0) vs. t(2)	Group A (*n* = 10)	*p*-Value t(0) vs. t(1) t(0)vs. t(2)	Group B (*n* = 7)	*p*-Value t(0) vs. t(1) t(0) vs. t(2)
Total insulin per day (IU)						
t(0)	138 ± 35		139 ± 34		136 ± 40	
t(1)	126 ± 42	*p =* 0.04	128 ± 46	*p* = 0.12	125 ± 37	*p* = 0.25
t(2)	127 ± 42	*p =* 0.05	135 ± 47	*p* = 0.47	117 ± 36	*p =* 0.04
Short-acting insulin per day (IU)						
t(0)	105 ± 37		107 ± 34		102 ± 43	
t(1)	95 ± 41	*p =* 0.03	101 ± 46	*p* = 0.14	88 ± 35	*p* = 0.15
t(2)	95 ± 45	*p =* 0.05	104 ± 50	*p* = 0.54	83 ± 37	*p =* 0.03
Long-acting insulin per day (IU)						
t(0)	36 ± 27		32 ± 21		41 ± 35	
t(1)	31 ± 29	*p* = 0.19	27 ± 20	*p* = 0.21	37 ± 39	*p* = 0.56
t(2)	32 ± 27	*p* = 0.20	31 ± 21	*p* = 0.36	34 ± 36	*p* = 0.31
HbA1 c (%)						
t(0)	8.1 ± 0.5		8.0 ± 0.5		8.1 ± 0.6	
t(1)	7.8 ± 0.6	*p =* 0.01	8.0 ± 0.6	*p* = 0.49	7.6 ± 0.6	*p =* 0.004
t(2)	8.1 ± 0.6	*p* = 0.71	8.0 ± 0.5	*p* = 1	8.2 ± 0.7	*p* = 0.61
HbA1 c (mmol/mol)						
t(0)	65.03 ± (−18.03)		63.93 ± (−16.94)		65.03 ± (−16.94)	
t(1)	61.75 ± (−16.94)	*p =* 0.01	63.93 ± (−16.94)	*p* = 0.49	59.56 ± (−16.94)	*p =* 0.004
t(2)	65.03 ± (−16.94)	*p* = 0.71	63.93 ± (−16.94)	*p* = 1	66.12 ± (−15.85)	*p* = 0.61
Body weight (kg)						
t(0)	103.9 ± 20.3		104.8 ± 18.2		102.5 ± 24.6	
t(1)	102.6 ± 20.2	*p =* 0.02	103.1 ± 17.7	*p =* 0.05	101.8 ± 24.8	*p* = 0.35
t(2)	102.6 ± 20.3	*p =* 0.04	104.6 ± 18.4	*p* = 0.63	99.7 ± 24.0	*p =* 0.02
BMI (kg/m^2^)						
t(0)	35.9 ± 5.5		35.6 ± 4.9		36.3 ± 6.7	
t(1)	35.5 ± 5.7	*p =* 0.04	35.0 ± 4.9	*p =* 0.05	36.1 ± 7.0	*p* = 0.44
t(2)	35.5 ± 5.7	*p =* 0.05	35.5 ± 5.1	*p* = 0.58	35.6 ± 6.8	*p =* 0.04
Systolic blood pressure (mmHg)						
t(0)	149 ± 20		146 ± 17		152 ± 25	
t(1)	146 ± 22	*p* = 0.57	149 ± 25	*p* = 0.71	142 ± 17	*p* = 0.31
t(2)	144 ± 24	*p* = 0.48	148 ± 29	*p* = 0.78	140 ± 16	*p* = 0.34
Diastolic blood pressure (mmHg)						
t(0)	85 ± 15		82 ± 11		86 ± 19	
t(1)	80 ± 16	*p* = 0.28	82 ± 13	*p* = 0.91	77 ± 21	*p* = 0.16
t(2)	80 ± 14	*p* = 0.24	83 ± 14	*p* = 0.65	75 ± 13	*p =* 0.05

Data are mean ± SD or *n* (%).

## Data Availability

The datasets generated during and/or analyzed during the current study are available from the corresponding author on reasonable request.

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
