# Peer review of "A Randomized Controlled Trial to Assess the Feasibility and Practicability of an Oatmeal Intervention in Individuals with Type 2 Diabetes: A Pilot Study in the Outpatient Sector"

_jcm, 2024, doi:10.3390/jcm13175126_

Round 1
Reviewer 1 Report
Comments and Suggestions for Authors
Dear Authors,
This is very interesting study , I have only a few questions/suggestions:
1. Title- Authors described study populations as metabolicalu unstable- but patients had no any unstability, they have rather only control of diabetes not in target.
2. Title- Will be worth mention that population is rather elderly
3. Inclusion criteria- If in the study intervention is calories restriction diet- I think in inclusion criteria should be high BMI- and mention about it in the study title.
3. Material and methods- Intervention- Please describe more precisely- what is the Oat preparation, how it was prepared- methods shoud be reproducible. Were these dietary recomendations individualized for patients with diferent body mass and activity?
4. Results- Biomedical outcomes- this part is not very good readible- the same data are in the table- maybe in this part shoud be only which results were statistically different after intervention.
Author Response
- Title: Authors described study populations as metabolically unstable- but patients had no any unstability, they have rather only control of diabetes not in target.
Answer: The term "metabolically unstable" referred to the correlation between the large amount of insulin, body weight and HbA1c. However, we have removed the term to avoid confusion. Thank you for your comment.
- Title: Will be worth mention that population is rather elderly.
Answer: That is correct. The average age of 70 years reflects the normal age of patients with type 2 diabetes. However, as this was not an inclusion criterion of our study, we would not want to mention the term "in elderly" in the title. We hope you agree with this.
- Inclusion criteria: If in the study intervention is calories restriction diet- I think in inclusion criteria should be high BMI- and mention about it in the study title.
Answer: Thank you for the opportunity to clarify this important aspect. The hypocaloric approach, which was investigated in this study using oat meals, was aimed at a possible improvement in insulin metabolism. The previous literature describes that three oat meals per day can lead to an improvement in insulin resistance. The calorie reduction was not the intervention to be investigated in the study, but resulted from the oat meals. Therefore, a high BMI was not an inclusion criterion of our study. The aim of our intervention was the interruption of insulin resistance, which was not defined by BMI.
- Material and methods- Intervention- Please describe more precisely- what is the Oat preparation, how it was prepared- methods should be reproducible. Were these dietary recommendations individualized for patients with different body mass and activity?
Answer: That is an important point. You can currently find the following description of the oat meal intervention in the manuscript: “At study entry, OI meal preparation was trained with the participants and recipes were handed out. Study participants were instructed to perform OI as follows: One oat-day in-cluded three self-prepared meals with 60g of rolled oats per portion. Different recipes for the two taste options sweet and savoury were given: The sweet version of oats (270 kcal, 41g carbohydrates (CHO), 9g protein, 7g fat; Table S1, appendix) was enriched with 200ml unsweetened almond drink (2.3% almonds) (i.e. 50g carbohydrates (CHO)). To make meals more palatable fruit according to personal preferences could be added containing 6g CHO, e.g. 50g apple or 30g banana or 100g strawberries. In the savoury version (261 kcal, 40g carbohydrates (CHO), 10g protein, 5g fat; Table S1, appendix), oats were pre-pared in 200ml of broth with 150g vegetables (containing 4.5g CHO) and flavored with herbs ad libitum (i.e. 40g CHO). Participants could choose which type of preparation they preferred. The total daily intake was 180g oats (containing 120g CHO) and approx. 800kcal (Table S1, appendix). During OI-days, no further food intake was allowed, unless for carbohydrates to counteract hypoglycemia. An adequate calorie-free fluid intake (1,5-2L/d) was recommended”.
These dietary recommendations were the same for all study patients regardless of body weight and activity level. This is due to the fact that we were investigating an intervention as described in the literature, we did not make any changes. We added this information to the manuscript. Thank you for that hint. In addition, sweet and savoury recipe was listed in the supplementary materials. We hope that is sufficient.
- Results- Biomedical outcomes- this part is not very good readable- the same data are in the table- maybe in this part should be only which results were statistically different after intervention.
Answer: Thank you very much for this helpful comment. We have revised this section. In Addition, we have simplified Figure 2 to avoid redundancy of HbA1c, body weight and blood pressure.
Reviewer 2 Report
Comments and Suggestions for Authors
Congratulations on your interesting and practical study. I have a few comments to improve the manuscript:
1. Please include the CONSORT checklist as supplementary material.
2. The study's primary endpoint is to evaluate feasibility and practicability. However, the feasibility-related results and discussion need to be added.
3. The discussion section should be improved to include all results related to the hypothesis.
4. Table 1 should include p-values.
5. Figure 2 needs to be clearer; currently, it is difficult to understand.
Author Response
- Please include the CONSORT checklist as supplementary material.
Answer: Thank you for this helpful comment. We added the CONSORT checklist and diagram (flow chart). Please find this information in our supplemental material.
- The study's primary endpoint is to evaluate feasibility and practicability. However, the feasibility-related results and discussion need to be added.
Answer: As a pilot study and proof-of-principle concept, feasibility was defined and assumed to be present if 70% and more of the study cohort had success-fully performed the intervention. Please find this information also in our manuscript (section methods 2.2 outcomes). We refer to this in the results as follows: “3.1. Feasibility of intervention: 13 out of 17 individuals (76.5%) completed the intervention period with no difference concerning the intervention arms (Group A: 7/10 participants (70.0%); Group B: 6/7 (85.7%); p=0.60). 2 out of 4 patients left the intervention due to OI-related reasons (unsatis-faction with OI [one], laboriousness for self-preparation at home [one]).”.Furthermore, we also address this point in the discussion (first section). Unfortunately, we do not have any additional aspects, as this is a pilot study. We hope that we have answered your justified comment sufficiently.
- The discussion section should be improved to include all results related to the hypothesis.
Answer: This study examines the feasibility and practicability of an oat based dietary intervention in the daily routine of outpatients with diabetes type 2 and insulin resistance. Additionally, we examined if OI can ameliorate IR and glycemic control in these patients. We have revised the discussion with regard to these questions. Thank you for your comment.
- Table 1 should include p-values.
Answer: Thank you for this aspect. We included the missing p-values. Please find the missing data in table 1.
- Figure 2 needs to be clearer; currently, it is difficult to understand.
Answer: Thank you for this helpful comment. We have made Figure 2 clearer and only referred to the insulin dose. We hope this is to your satisfaction.
Reviewer 3 Report
Comments and Suggestions for Authors
The manuscript " A randomized controlled trial to assess feasibility and practica- 2 bility of an oatmeal intervention in metabolically unstable in- 3 dividuals with type 2 diabetes: a pilot study in the outpatient 4 sector " is an interesting a pilot study in the outpatient sector, health-related study however there are some inconsistencies.
Abstract: The abstract is appropriate and complete.
Introduction section: The introduction section is appropriate and complete.
Results
In table 3. I suggest using another statistical test since it repeats the Student's t-test comparing time 0 with time 1, as well as comparing time 0 vs time 2. This procedure generates the type 2 error in the statistics, so if possible another way of analysing the data should be sought.
Since you have a small sample size, it affects your decision on which statistical test to run. Evaluate the normality of your data used in the study.
2. Materials and Methods
2.1 Intervention
I suggest adding the caloric and percentage distribution of the other macronutrients (protein and lipids), as this could be the reason why the patients in your study did not lose enough weight.
References
The references are appropriate and complete.
Author Response
- Abstract: The abstract is appropriate and complete.
Answer: Thank you for your comment!
- Introduction section: The introduction section is appropriate and complete.
Answer: Thank you very much.
- Results: In table 3. I suggest using another statistical test since it repeats the Student's t-test comparing time 0 with time 1, as well as comparing time 0 vs time 2. This procedure generates the type 2 error in the statistics, so if possible another way of analysing the data should be sought. Since you have a small sample size, it affects your decision on which statistical test to run. Evaluate the normality of your data used in the study.
Answer: Thank you also for this comment, which we would like to seize on. We agree with you completely that our sample size is small. We discussed this point with the “Institute for Medical Statistics, Informatics and Data Science (IMSID)” of the University Hospital Jena in advance and we have done so repeatedly. In consultation with our statistician, we have selected the tests described in the manuscript and we hope you agree with them.
- Materials and Methods, 2.1 Intervention: I suggest adding the caloric and percentage distribution of the other macronutrients (protein and lipids), as this could be the reason why the patients in your study did not lose enough weight.
Answer: Thank you for that hint. We added the information to the section “2.1 Intervention” of the methods and referred to Table 1 of the appendix.
- References: The references are appropriate and complete.
Answer: Thank you for your comment!